# LEARNED INDEX WITH DYNAMIC $\epsilon$

## ABSTRACT

Index structure is a fundamental component in database and facilitates broad data retrieval applications. Recent learned index methods show superior performance by learning hidden yet useful data distribution with the help of machine learning, and provide a guarantee that the prediction error is no more than a pre-defined $\epsilon$. However, existing learned index methods adopt a fixed $\epsilon$ for all the learned segments, neglecting the diverse characteristics of different data localities. In this paper, we propose a mathematically-grounded learned index framework with dynamic $\epsilon$, which is efficient and pluggable to existing learned index methods. We theoretically analyze prediction error bounds that link $\epsilon$ with data characteristics for an illustrative learned index method. Under the guidance of the derived bounds, we learn how to vary $\epsilon$ and improve the index performance with a better space-time trade-off. Experiments with real-world datasets and several state-of-the-art methods demonstrate the efficiency, effectiveness and usability of the proposed framework.

## 1 INTRODUCTION

Data indexing (Graefe & Kuno, 2011; Wang et al., 2018; Luo & Carey, 2020; Zhou et al., 2020), which stores keys and corresponding payloads with designed structures, supports efficient query operations over data and benefits various data retrieval applications. Recently, Machine Learning (ML) models have been incorporated into the design of index structure, leading to substantial improvements in terms of both storage space and querying efficiency (Kipf et al., 2019; Ferragina & Vinciguerra, 2020a; Mitzenmacher, 2018; Vaidya et al., 2021). The key insight behind this trending topic of "learned index" is that the data to be indexed contain useful distribution information and such information can be utilized by trainable ML models that map the keys to their stored positions. State-of-the-art learned index methods (Galakatos et al., 2019; Kipf et al., 2020; Ferragina & Vinciguerra, 2020b; Ferragina et al., 2020) adopt piece-wise linear segments to approximate the data distribution and introduce an important pre-defined parameter $\epsilon$. These methods ensure that the maximal prediction error of each learned segment is no more than $\epsilon$ and provide a worst-case guarantee of querying efficiency.

By tuning $\epsilon$, various space-time preferences from users can be met. For example, a relatively large $\epsilon$ can result in a small index size while having large prediction errors, and on the other hand, a relatively small $\epsilon$ provides with small prediction errors while having more learned segments and thus a large index size. However, existing learned index methods implicitly assume that the whole dataset to be indexed contains the same characteristics for different localities and thus adopt the same $\epsilon$ for all the learned segments, leading to sub-optimal index performance. More importantly, the impact of $\epsilon$ on index performance is intrinsically linked to data characteristics, which are not fully explored and utilized by existing learned index methods.

Motivated by these, in this paper, we theoretically analyze the impact of $\epsilon$ on index performance, and link the characteristics of data localities with the dynamic adjustments of $\epsilon$. Based on the derived theoretical results, we propose an efficient and pluggable learned index framework that dynamically adjusts $\epsilon$ in a principled way. To be specific, under the setting of an illustrative learned index method MET (Ferragina et al., 2020), we present novel analysis about the prediction error bounds of each segment that link $\epsilon$ with the mean and variance of data localities. The segment-wise prediction error embeds the space-error trade-off as it is the product of the *number of covered keys* and *mean absolute error*, which determine the index size and preciseness respectively. The derived mathematical relationships enable our framework to fully explore diverse data localities with an $\epsilon$-learner module, which learns to predict the impact of $\epsilon$ on the index performance and adaptively choose a suitable $\epsilon$ to achieve a better space-time trade-off.

We apply the proposed framework to several state-of-the-art (SOTA) learned index methods, and conduct a series of experiments on three widely adopted real-world datasets. Comparing with the original learned index methods with fixed $\epsilon$, our dynamic $\epsilon$ versions achieve significant index performance improvements with better space-time trade-offs. We also conduct various experiments to verify the necessity and effectiveness of the proposed framework, and provide both ablation study and case study to understand how the proposed framework works.

## 2 BACKGROUND

### 2.1 $\epsilon$-BOUNDED LEARNED INDEX

Given a dataset $\mathcal{D} = \{(x, y) | x \in \mathcal{X}, y \in \mathcal{Y}\}$, $\mathcal{X}$ is the set of *keys* over a universe $\mathcal{U}$ such as reals or integers, and $\mathcal{Y}$ is the set of *positions* where the keys and corresponding payloads are stored. The index such as B$^+$-tree (Abel, 1984) aims to build a compact structure to support efficient query operations over $\mathcal{D}$. Typically, the keys are assumed to be sorted in ascending order to satisfy the *key-position monotonicity*, *i.e.*, for any two keys, $x_i > x_j$ iff their positions $y_i > y_j$, such that the range query ($\mathcal{X} \cap [x_{low}, x_{high}]$) can be handled.

Recently, learned index methods (Kraska et al., 2018; Li et al., 2019; Tang et al., 2020; Dai et al., 2020; Crotty, 2021) leverage ML models to mine useful distribution information from $\mathcal{D}$, and incorporate such information to boost the index performance. To look up a given key $x$, the learned index first predicts position $\hat{y}$ using the learned models, and subsequently finds the stored true position $y$ based on $\hat{y}$ with a binary search or exponential search. Thus the querying time consists of the inference time of the learned models and the search time in $O(\log(|\hat{y} - y|))$. By modeling the data distribution information, learned indexes achieve faster query speed than traditional B$^+$-tree index, meanwhile using several orders-of-magnitude smaller storage space (Ding et al., 2020; Galakatos et al., 2019; Ferragina & Vinciguerra, 2020b; Kipf et al., 2020; Marcus et al., 2020).

Many existing learned index methods adopt piece-wise linear segments to approximate the distribution of $\mathcal{D}$ due to their effectiveness and low computing cost, and introduce the parameter $\epsilon$ to provides a worst-case preciseness guarantee and a tunable knob to meet various space-time trade-off preferences. Here we briefly introduce the SOTA $\epsilon$-bounded learned index methods that are most closely to our work, and refer to the review chapter of (Ferragina & Vinciguerra, 2020a) for details of other methods. Specifically, Galakatos et al. (2019) greedily learn a set of piece-wise linear segments in a one-pass manner. Ferragina & Vinciguerra (2020b) adopt another one-pass algorithm that achieves the optimal number of learned segments. Kipf et al. (2020) introduce a radix structure to organize the learned segments. Ferragina et al. (2020) adopt a fixed slope setting for all learned segments. Existing methods constraint all learned segments with the same $\epsilon$, *i.e.*, the learning process ensures that the maximum prediction error is within a pre-defined $\epsilon$ where $\epsilon \in \mathbb{Z}_{>1}$. In this paper, we will discuss the impact of $\epsilon$ in more depth and invistigate how to enhance existing learned index methods from a new perspective: dynamic adjustment of $\epsilon$ considering the diversity of different data localities.

### 2.2 MEAN EXIT TIME (MET) ALGORITHM

Here we describe an illustrative learned index algorithm MET (Ferragina et al., 2020). Specifically, for any two consecutive keys of $\mathcal{D}$, suppose their key interval ($x_i - x_{i-1}$) is drawn according to a random process $\{G_i\}_{i \in \mathbb{N}}$, where $G_i$ is a positive independent and identically distributed (i.i.d.) random variable whose mean is $\mu$ and variance is $\sigma^2$. The MET algorithm learns linear segments $\mathbf{S} = [S_1, ..., S_i, ..., S_N]$ in one pass, where $S_i : y = a_i x + b_i$ is the learnable linear segment and $N$ is the total number of learned segments. The learning process is as follows: The current linear segment fixes the slope $a_i = 1/\mu$ and goes through the first available data point, thus $b_i$ is also determined. Then the segment covers as many data points as possible until a data point, say $(x', y')$ achieves the prediction error larger than $\epsilon$. The violation of $\epsilon$ triggers a new linear segment, and the data point $(x', y')$ will be the first available data point, and the process repeats until no data point is available. Although the MET algorithm adopts a simple learning mechanism, it makes the theoretical analysis convenient by modeling the learning process as a random walk process. Other $\epsilon$-bounded learned index methods such as FITing-Tree (Galakatos et al., 2019), PGM (Ferragina & Vinciguerra, 2020b) and Radix-Spline (Kipf et al., 2020) learn linear segments in a similar manner while having different mechanisms to determine the parameters of $\{S_i\}$. Ferragina et al. (2020) reveal the relationship

between $\epsilon$ and index size performance based on MET. In Section 3.3, we present novel analysis about the impact of $\epsilon$ on not only the index size, *but also the index preciseness and a comprehensive trade-off quantity*, which facilitates the proposed dynamic $\epsilon$ adjustment.

# 3 LEARN TO VARY $\epsilon$

## 3.1 PROBLEM FORMULATION AND MOTIVATION

Before introducing the proposed framework, we first formulate the task of learning index from data with $\epsilon$ guarantee, and provide some discussions about why we need to vary $\epsilon$. Given a dataset $\mathcal{D}$ to be indexed and an $\epsilon$-bounded learned index algorithm $\mathcal{A}$, we aim to learn segments $\mathbf{S} = [S_1, ..., S_i..., S_N]$ as index structure such that the size of $\mathbf{S}$ and $\sum_{i=1}^{N} MAE(\mathcal{D}_i|S_i)$ are both small, where $S_i$ is the $i$-th learned linear segment with the parameter $\epsilon_i$, *MAE* is the mean absolute prediction error, and $\mathcal{D}_i \subset \mathcal{D}$ is the data whose keys are covered by $S_i$. For the remaining data $\mathcal{D} \setminus \bigcup_{j<i} \mathcal{D}_j$, the algorithm $\mathcal{A}$ repeatedly checks whether the prediction error of new data point violates the given $\epsilon_i$, and outputs the learned segment $S_i$ that covers $\mathcal{D}_i$. When all the $\epsilon_i$s for $i \in [N]$ take the same value, the problem becomes the one that existing learned index methods are dealing with.

Now let's examine the effect of parameter $\epsilon$. To query a specific data point, say $(x, y)$, we first need to find the specific segment $S'$ that covers $x$, and then search its true position $y$ based on the estimated one $\hat{y} = S'(x)$. For the first step, we can find $S'$ from $\mathbf{S}$ in $O(\log(N))$; for the second step, the search of $y$ based on $\hat{y}$ can be done in $O(\log(|\hat{y} - y|))$. In summary, we can find the true position of the queried data point in $O(\log(N) + \log(|\hat{y} - y|))$ [1]. From here, we can see that the parameter $\epsilon$ plays an important role to trade off two contradictory performance terms, *i.e.*, the size of the learned index $N$, and *MAE* of the whole data $MAE(\mathcal{D}|\mathbf{S})$. If we adopt a small $\epsilon$, the maximal prediction error constraint is more frequently violated, leading to a large $N$; meanwhile, the preciseness of learned index is improved, leading to a small $MAE(\mathcal{D}|\mathbf{S})$. On the other hand, with a large $\epsilon$, we will get a more compact learned index (*i.e.*, a small $N$) with larger prediction errors (*i.e.*, a large $MAE(\mathcal{D}|\mathbf{S})$). Thus these two inversely changed terms jointly impact the querying efficiency.

Actually, the effect of $\epsilon$ on index performance is intrinsically linked to the characteristic of the data to be indexed. For real-world datasets, an important observation is that *the linearity degree varies in different data localities*. Recall that we use piece-wise linear segments to fit the data, and $\epsilon$ determines the partition and the fitness of the segments. By varying $\epsilon$, we can adapt to the local variations of $\mathcal{D}$ and adjust the partition such that each learned segment fits the data better. Formally, let's consider the quantity $SegErr_i$ that is defined as the total prediction error within a segment $S_i$, *i.e.*, $SegErr_i = \sum_{(x,y) \in \mathcal{D}_i} |y - S_i(x)|$, which is also the product of the number of covered keys $Len(\mathcal{D}_i)$ and the mean absolute error $MAE(\mathcal{D}_i|S_i)$. Note that a large $Len(\mathcal{D}_i)$ leads to a small $N$ since $|\mathcal{D}| = \sum_{i=1}^{N} Len(\mathcal{D}_i)$. From this view, the quantity $SegErr_i$ internally reflects the space-error trade-off. Later we will show how to leverage this quantity to dynamically adjust $\epsilon$.

## 3.2 OVERALL FRAMEWORK

In practice, it is intractable to directly solve the problem formulated in Section 3.1. With a given $\epsilon_i$, the one-pass algorithm $\mathcal{A}$ determines $S_i$ and $\mathcal{D}_i$ until the error bound $\epsilon_i$ is violated. In other words, it is unknown what the data partition $\{\mathcal{D}_i\}$ will be *a priori*, which makes it impossible to solve the problem by searching among all the possible $\{\epsilon_i\}$s and learning index with a set of given $\{\epsilon_i\}$.

In this paper, we investigate how to efficiently find an approximate solution to this problem via the introduced $\epsilon$-*learner* module. Instead of heuristically adjusting $\epsilon$, the $\epsilon$-*learner* learns to predict the impact of $\epsilon$ on the index structure and adaptively adjusts $\epsilon$ in a principled way. Meanwhile, the introducing of $\epsilon$-*learner* should not sacrifice the efficiency of the original one-pass learned index algorithms, which is important for real-world practical applications.

These two design considerations establish our dynamic $\epsilon$ framework as shown in Figure 1. The $\epsilon$-*learner* is based on an estimation function $SegErr = f(\epsilon, \mu, \sigma)$ that depicts the mathematical relationships among $\epsilon$, $SegErr_i$ and the characteristics $\mu, \sigma$ of the data to be indexed. As a start, users can provide an expected $\tilde{\epsilon}$ that indicates various preferences under space-sensitive or time-sensitive applications. To meet the user requirements, afterwards, we will internally transform the $\tilde{\epsilon}$ into another

---

[1] In Appendix C, we link the absolute prediction error and specific searching algorithms in further details.

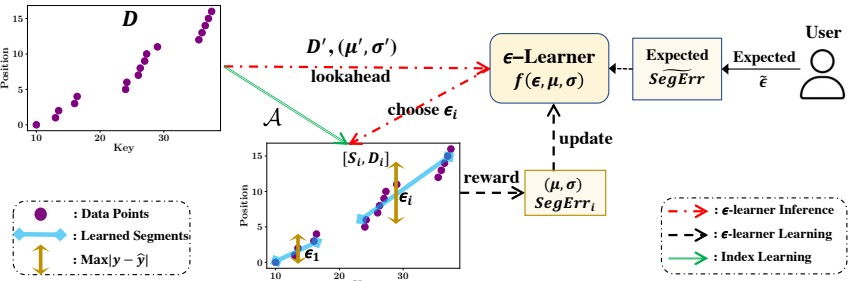

Figure 1: Dynamic $\epsilon$ framework with the $\epsilon$-*learner* module.

proxy quantity $\widetilde{SegErr}$, which reflects the expected prediction error for each segment if we set $\epsilon_i = \tilde{\epsilon}$. This transformation also links the adjustment of $\epsilon$ and data characteristics together, which enables the data-dependent adjustment of $\epsilon$. Beginning with $\tilde{\epsilon}$, the $\epsilon$-*learner* chooses a suitable $\epsilon_i$ according to current data characteristics, then learn a segment $S_i$ using $\mathcal{A}$, and finally enhance the $\epsilon$-*learner* with the rewarded ground-truth $SegErr_i$ of each segment. To make the introduced adjustment efficient, we propose to only sample a small *Look-ahead* data $\mathcal{D}'$ to estimate the characteristics (*i.e.*, $\mu$ and $\sigma$) of the following data locality. The learning and adjusting processes are repeatedly conducted and also in an efficient one-pass manner.

Note that the proposed framework provides users the same interface as the ones used by original learned index methods. That is, we add no any additional cost to the users' experience, and users can smoothly and painlessly use our framework with given $\tilde{\epsilon}$ just as they use the original methods with given $\epsilon$. The $\epsilon$ is an intuitive, easy-to-set and method-agnostic quantity for users. On the one hand, we can easily impose restrictions on the worst-case querying cases with $\epsilon$ as the data accessing number in querying process is $O(\log(|\hat{y} - y|))$. On the other hand, $\epsilon$ is easier to estimate than the other quantities such as index size and querying time, which are dependent on specific algorithms, data layouts, implementations and experimental platforms. Our pluggable framework retains the benefits of existing learned index methods, such as the aforementioned usability of $\epsilon$ and the ability to handle dynamic update case (Galakatos et al., 2019; Ferragina & Vinciguerra, 2020b).

We have seen how $\epsilon$ determines index performance and how $SegErr_i$ embeds the space-error trade-off in Section 3.1. In Section 3.3, we will further theoretically analyze the relationship among $\epsilon$, $SegErr_i$, and data characteristics $\mu, \sigma$ at different localities. Based on the analysis, we elaborate the details of $\epsilon$-*learner* and the internal transformation between $\epsilon$ and $SegErr_i$ in Section 3.4.

### 3.3 PREDICTION ERROR ESTIMATION

In this section, we will theoretically study the impact of $\epsilon$ on the prediction error $SegErr_i$ of each learned segment $S_i$. Specifically, for the MET algorithm, we can prove the following theorem to bound the expectation of $SegErr_i$ with $\epsilon$ and the key interval distribution of the data to be indexed.

**Theorem 1.** *Given a dataset $\mathcal{D}$ to be indexed and an $\epsilon$ where $\epsilon \in \mathbb{Z}_{>1}$, consider the setting of the MET algorithm (Ferragina et al., 2020), in which key intervals of $\mathcal{D}$ are drawn from a random process consisting of positive i.i.d. random variables with mean $\mu$ and variance $\sigma^2$, and $\epsilon \gg \sigma/\mu$. For a learned segment $S_i$ and its covered data $\mathcal{D}_i$, denote $SegErr_i = \sum_{(x,y) \in D_i} |y - S_i(x)|$. Then the expectation of $SegErr_i$ satisfies:*

$$\sqrt{\frac{1}{\pi}\frac{\mu}{\sigma}}\epsilon^2 < \mathbb{E}[SegErr_i] < \frac{2}{3}\sqrt{\frac{2}{\pi}}(\frac{5}{3})^{\frac{3}{4}}(\frac{\mu}{\sigma})^2\epsilon^3.$$

This theorem reveals that the prediction error $SegErr_i$ depends on both $\epsilon$ and the data characteristics ($\mu$ and $\sigma$). Recall that $CV = \sigma/\mu$ is the *coefficient of variation*, a classical statistical measure of the relative dispersion of data points. In the context of the linear approximation, the data statistic $1/CV = \mu/\sigma$ in our bounds intrinsically corresponds to the linearity degree of the data. With this, we can find that when $\mu/\sigma$ is large, the data is easy-to-fit with linear segments, and thus we can choose a small $\epsilon$ to achieve precise predictions. On the other hand, when $\mu/\sigma$ is small, it becomes harder to fit the data using a linear segment, and thus $\epsilon$ should be increased to absorb some non-linear data localities. In this way, we can make the total prediction error $SegErr_i$ for different learned segments consistent and achieve a better space-error trade-off. This analysis also confirms the motivation of

varying $\epsilon$: The local linearity degrees of the indexed data can be diverse, and we should adjust $\epsilon$ according to the local characteristic of the data, such that the learned index can fit and leverage the data distribution better. In the design of the $\epsilon$-learner module (Section 3.4), we will take the derived closed-form relationships among $SegErr_i$, $\epsilon$ and data statistic $\mu/\sigma$ into account.

In the rest of this section, we provide a proof sketch of this theorem due to the space limitation. For detailed proof, please refer to our Appendix. The main idea is to model the learning process of linear approximation with $\epsilon$ guarantee as a random walk process, and consider that the absolute prediction error of each data point follows folded normal distributions. Specifically, given a learned segment $S_i : y = a_i x + b_i$, we can calculate the expectation of $SegErr_i$ for this segment as:

$$\mathbb{E}[SegErr_i] = a_i \mathbb{E}\left[\sum_{j=0}^{(j^*-1)} |Z_j|\right] = a_i \sum_{n=1}^{\infty} \mathbb{E}\left[\sum_{j=0}^{n-1} |Z_j|\right] \Pr(j^* = n), \qquad (1)$$

where $Z_j$ is the $j$-th position of a transformed random walk $\{Z_j\}_{j\in\mathbb{N}}$, $j^* = \max\{j \in \mathbb{N}| -\epsilon/a_i \leq Z_j \leq \epsilon/a_i\}$ is the random variable indicating the maximal position when the random walk is within the strip of boundary $\pm\epsilon/a_i$, and the last equality is due to the definition of expectation.

Under the MET algorithm setting where $a_i = 1/\mu$ and $\epsilon \gg \sigma/\mu$, we can show that the increments of the transformed random walk $\{Z_j\}$ have zero mean and variance $\sigma^2$, and many steps are necessary to reach the random walk boundary. With the Central Limit Theorem, we can assume the $Z_j$ follows normal distribution with mean $\mu_{zj} = 0$ and variance $\sigma_{zj}^2 = j\sigma^2$, and thus $|Z_j|$ follows the folded normal distribution with expectation $\mathbb{E}(|Z_j|) = \sqrt{2/\pi}\sigma\sqrt{j}$. Thus Eq. (1) can be written as

$$\frac{1}{\mu}\sum_{n=1}^{\infty}\mathbb{E}\left[\sum_{j=0}^{n-1}|Z_j|\right]\Pr(j^* = n) < \frac{1}{\mu}\sum_{n=1}^{\infty}\sum_{j=0}^{n-1}\mathbb{E}[|Z_j|]\Pr(j^* = n) = \frac{\sigma}{\mu}\sqrt{\frac{2}{\pi}}\sum_{n=1}^{\infty}\sum_{j=0}^{n-1}\sqrt{j}\Pr(j^* = n).$$

Using $\mathbb{E}[j^*] = \frac{\mu^2}{\sigma^2}\epsilon^2$ and $Var[j^*] = \frac{2}{3}\frac{\mu^4}{\sigma^4}\epsilon^4$ as derived in (Ferragina et al., 2020), we get $\mathbb{E}[(j^*)^2] = \frac{5}{3}\frac{\mu^4}{\sigma^4}\epsilon^4$. With the inequality $\sum_{j=0}^{n-1}\sqrt{j} < \frac{2}{3}n\sqrt{n}$ and $\mathbb{E}[X^{\frac{3}{4}}] \leq (\mathbb{E}[X])^{\frac{3}{4}}$, we get the upper bound:

$$\mathbb{E}[SegErr_i] < \frac{2}{3}\sqrt{\frac{2}{\pi}}\frac{\sigma}{\mu}\mathbb{E}[(j^*)^{\frac{3}{2}}] \leq \frac{2}{3}\sqrt{\frac{2}{\pi}}\frac{\sigma}{\mu}\left(\mathbb{E}[(j^*)^2]\right)^{\frac{3}{4}} = \frac{2}{3}\sqrt{\frac{2}{\pi}}(\frac{5}{3})^{\frac{3}{4}}(\frac{\mu}{\sigma})^2\epsilon^3.$$

For the lower bound, applying the triangle inequality into Eq. (1), we can get $\mathbb{E}[SegErr_i] > \frac{1}{\mu}\sum_{n=1}^{\infty}\mathbb{E}[|Z|]\Pr(j^* = n)$, where $Z = \sum_{j=0}^{n-1}Z_j$, and $Z$ follows the normal distribution since $Z_j \sim N(0, \sigma_{zj}^2)$. We can prove that $|Z|$ follows the folded normal distribution whose expectation $\mathbb{E}[|Z|] > \sigma(n-1)/\sqrt{\pi}$. Thus the lower bound is:

$$\mathbb{E}[SegErr_i] > \frac{\sigma}{\mu}\sqrt{\frac{1}{\pi}}\sum_{n=1}^{\infty}(n-1)\Pr(j^* = n) = \frac{\sigma}{\mu}\sqrt{\frac{1}{\pi}}\mathbb{E}[j^* - 1] = \sqrt{\frac{1}{\pi}}(\frac{\mu}{\sigma}\epsilon^2 - \frac{\sigma}{\mu}).$$

Since $\epsilon \gg \frac{\sigma}{\mu}$, we can omit the right term $\sqrt{1/\pi} \cdot \sigma/\mu$ and finish the proof. Although the derivations are based on the MET algorithm whose slope is the reciprocal of $\mu$, we found that the mathematical forms among $\epsilon$, $\mu/\sigma$ and $SegErr_i$ are still applicable to other $\epsilon$-bounded methods, and further prove that the learned segment slopes of other methods are close to the reciprocal of expected key intervals in Appendix. We will empirically show the links between MET and other SOTA $\epsilon$-bounded methods, and how effectively the proposed framework works for them on real-world datasets (Section 4.2).

## 3.4 $\epsilon$-LEARNER

Now given an $\epsilon$, we have obtained the closed-form bounds of the $SegErr$ in Theorem 1, and both the upper and lower bounds are in the form of $w_1(\frac{\mu}{\sigma})^{w_2}\epsilon^{w_3}$, where $w_{1,2,3}$ are some coefficients. As the concrete values of these coefficients can be different for different datasets and different methods, we propose to learn the following trainable estimator to make the error prediction preciser:

$$SegErr = f(\epsilon, \mu, \sigma) = w_1(\frac{\mu}{\sigma})^{w_2}\epsilon^{w_3},$$

$$s.t. \qquad \sqrt{\frac{1}{\pi}} \leq w_1 \leq \frac{2}{3}\sqrt{\frac{2}{\pi}}(\frac{5}{3})^{\frac{3}{4}}, \quad 1 \leq w_2 \leq 2, \quad 2 \leq w_3 \leq 3. \qquad (2)$$

With this learnable estimator, we feed data characteristic $\mu/\sigma$ of the look-ahead data and the transformed $\widetilde{SegErr}$ into it and find a suitable $\epsilon^*$ as $\left( \widetilde{SegErr}/w_1(\frac{\mu}{\sigma})^{w_2} \right)^{1/w_3}$. We will discuss the look-ahead data and the transformed $\widetilde{SegErr}$ in the following paragraphs. Now let's discuss the reasons for how this adjustment can achieve better index performance. Actually, the $\epsilon$-learner proactively plans the allocations of the total prediction error indicated by user (*i.e.*, $\tilde{\epsilon} \cdot |\mathcal{D}|$) and calculates the tolerated $\widetilde{SegErr}$ for the next segment. By adjusting current $\epsilon$ to $\epsilon^*$, the following learned segment can fully utilize the distribution information of the data and achieve better performance in terms of space-error trade-off. To be specific, when $\mu/\sigma$ is large, the local data has clear linearity, and thus we can adjust $\epsilon$ to a relatively small value to gain precise predictions; although the number of data points covered by this segment may decrease and then the number of total segments increases, such cost paid in terms of space is not larger than the benefit we gain in terms of precise predictions. Similarly, when $\mu/\sigma$ is small, $\epsilon$ should be adjusted to a relatively large value to lower the learning difficulty and absorb some non-linear data localities; in this case, we gain in terms of space while paying some costs in terms of prediction accuracy. The segment-wise adjustment of $\epsilon$ improves the overall index performance by continually and data-dependently balancing the cost of space and preciseness.

**Look-ahead Data.** To make the training and inference of the $\epsilon$-learner light-weight, we propose to look ahead a few data $\mathcal{D}'$ to reflect the characteristics of the following data localities. Specifically, we leverage a small subset $\mathcal{D}' \subset \mathcal{D} \setminus \bigcup_{j<i} \mathcal{D}_j$ to estimate the value $\mu/\sigma$ for the following data. In practice, we set the size of $\mathcal{D}'$ to be $404$ when learning the first segment as initialization, and $\left( \frac{1}{(i-1)} \sum_{j=1}^{i-1} Len(\mathcal{D}_j) \right) \cdot \rho$ for the other following segments. Here $\rho$ is a pre-defined parameter indicating the percentage that is relative to the average number of covered keys for learned segments, considering that the distribution of $\mu/\sigma$ can be quite different to various datasets. As for the first segment, according to the literature (Kelley, 2007), the sample size $404$ can provide a 90% confidence intervals for a coefficient of variance $\sigma/\mu \leq 0.2$.

$\widetilde{SegErr}$ **and Optimization.** As aforementioned, taking the user-expected $\tilde{\epsilon}$ as input, we aim to reflect the impact of $\tilde{\epsilon}$ with a transformed proxy quantity $\widetilde{SegErr}$ such that the $\epsilon$-learner can choose suitable $\epsilon^*$ to meet users' preference while achieving better space-error trade-off. Specifically, we make the value of $\widetilde{SegErr}$ updatable, and update it to be $\widetilde{SegErr} = w_1(\hat{\mu}/\hat{\sigma})^{w_2}\tilde{\epsilon}^{w_3}$ once a new segment is learned, where $\hat{\mu}/\hat{\sigma}$ is the mean value of all the processed data so far. This strategy enables us to promptly incorporate both the user preference and the data distribution into the calculation of $\widetilde{SegErr}$. As for the optimization of the $f(\epsilon, \mu, \sigma)$, we adopt the projected gradient descent (Calamai & Moré, 1987; den Hertog & Roos, 1991) with the parameter constraints in Eq. (2). In this way, we only need to track a few statistics and learn the $\epsilon$ estimator in an efficient one-pass manner. The overall adjustment algorithm is summarized in Appendix D.

## 4 EXPERIMENTS

### 4.1 EXPERIMENTAL SETTINGS

**Baselines.** We apply our framework into several SOTA $\epsilon$-bounded learned index methods that use different mechanisms to determine the parameters of segments $\{S_i\}$. Among them, *MET* (Ferragina et al., 2020) fixes the segment slope as the reciprocal of the expected key interval. *FITing-Tree* (Galakatos et al., 2019) and *Radix-Spline* (Kipf et al., 2020) adopt a greedy shrinking cone algorithm and a spline interpolating algorithm respectively. *PGM* (Ferragina & Vinciguerra, 2020b) adopts a convex hull based algorithm to achieve the minimum number of learned segments. Further introduction and implementation details can be found in Appendix.

**Datasets.** We use several widely adopted datasets with differing data scales and distributions (Kraska et al., 2018; Galakatos et al., 2019; Ferragina & Vinciguerra, 2020b; Li et al., 2021). *Weblogs* and *IoT* contain about 715M log entries from a university web server and 26M event entries from different IoT sensors respectively, in which the keys are both log timestamps. *Map* dataset contains location coordinates that are collected around the world from the OpenStreetMap contributors (2017), and the keys are $longitude + 90 \cdot latitude$ of about 1.8M places. *Lognormal* is a synthetic dataset whose key intervals follow the lognormal distribution. We generate 20M keys with 40 partitions

having different generation parameters to simulate the varied data characteristics among different localities. More dataset details and visualization are presented in Appendix F.

**Evaluation Metrics.** We evaluate the index performance in terms of its size, prediction preciseness, and the total querying time. Specifically, we report the number of learned segments $N$, the index size in bytes, the *MAE* as $\frac{1}{|D|} \sum_{(x,y) \in D} |y - \mathbf{S}(x)|$, and the total querying time per query in ns (*i.e.*, we perform querying operations for all the indexed data, record the total time of getting the payloads given the keys, and report the time that is averaged over all the queries). For a quantitative comparison w.r.t. the trade-off improvements, we calculate the area under the space-error curve (AUSEC) where the x-axis and y-axis indicate $N$ and *MAE* respectively. For AUSEC metric, the smaller, the better.

## 4.2 OVERALL INDEX PERFORMANCE

**Space-Error Trade-off Improvements.** In Table 1, we summarize the AUSEC improvements in percentage brought by the proposed framework of all the baseline methods on all the datasets. We also illustrate the space-error trade-off curves for some cases in Figure 2, where the blue curves indicate the results achieved by fixed $\epsilon$ version while the red curves are for dynamic $\epsilon$. Other baselines and datasets yield similar curves, which we include in Appendix due to the space limitation. From these results, we can see that the dynamic $\epsilon$ versions of all the baseline methods achieve much better error-space trade-off ($-16.48\%$ to $-23.57\%$ averaged improvements as smaller AUSEC indicates better performance), demonstrating the effectiveness and the wide applicability of the proposed framework. As discussed in previous sections, datasets usually have diverse key distributions at different data localities, and the proposed framework can data-dependently adjust $\epsilon$ to fully utilize the distribution information of data localities and thus achieve better index performance in terms of space-error trade-off. Note that the Map dataset has significant non-linearity caused by spatial characteristics, and it is hard to fit using linear segments (all baseline methods learn linear segments), thus relatively small improvements are achieved.

Table 1: The AUSEC relative improvements for learned index methods with dynamic $\epsilon$.

|  | Weblogs | IoT | Map | Lognormal | Average |
|---|---|---|---|---|---|
| MET | -25.87% | -7.66% | -10.89% | -21.48% | -16.48% |
| FITing-Tree | -31.18% | -25.56% | -9.30% | -28.24% | -23.57% |
| Radix-Spline | -28.37% | -24.59% | -8.77% | -31.32% | -23.26% |
| PGM | -22.42% | -25.01% | -7.18% | -19.58% | -18.55% |

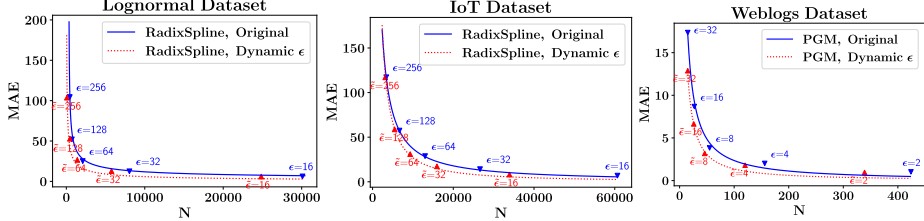

Figure 2: The space-error trade-off curves for learned index methods.

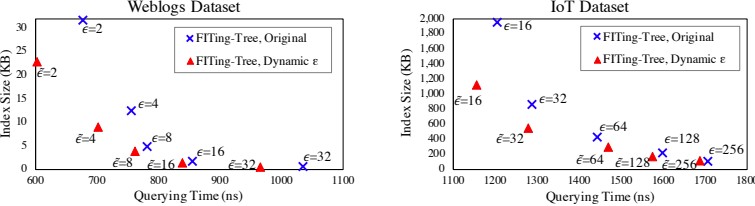

Figure 3: Improvements in terms of querying time for learned index methods with dynamic $\epsilon$.

**Querying Time Improvements.** Recall that the querying time of each data point is in $O(\log(N) + \log(|y - \hat{y}|))$ as we mentioned in Section 3.1, where $N$ and $|y - \hat{y}|$ are inversely impacted by $\epsilon$. To examine whether the performance improvements w.r.t. space-error trade-off (*i.e.*, Table 1) can lead to better querying efficiency in real-world systems, we show the averaged total querying time per

query and the actual learned index size in bytes for two scenarios in Figure 3. We can observe that the dynamic $\epsilon$ versions indeed gain faster querying speed, since we improve both the term $N$ as well as the term $|y - \hat{y}|$ via adaptive adjustment of $\epsilon$. The similar conclusion can be drawn from other baselines and datasets, and we present their results in Appendix. Another thing to note is that, this experiment also verifies the usability of our framework in which users can flexibly set the expected $\tilde{\epsilon}$ to meet various space-time preferences just as they set $\epsilon$ in the original learned index methods.

**Index Building Cost.** Comparing with the original learned index methods that adopt a fixed $\epsilon$, our framework introduces extra computation to dynamically adjust $\epsilon$ in the index building stage. Does this affect the efficiency of original learned index methods? Here we report the relative increments of building times in Table 2. From it, we can observe that the proposed dynamic $\epsilon$ framework achieves comparable building times to all the original learned index methods on all the datasets, showing the efficiency of our framework since it has the same complexity as the original methods (both in $O(|\mathcal{D}|)$). Also note that we only need to pay this extra cost once, *i.e.*, building the index once, and then the index structures can accelerate the frequent data querying operations for real-world applications.

Table 2: Building time increments in percentage for learned index methods with dynamic $\epsilon$.

|  | Weblogs | IoT | Map | Lognormal | Average |
|---|---|---|---|---|---|
| MET | 10.54% | 5.14% | 7.55% | 5.26% | 7.12% |
| FITing-Tree | 10.7% | 1.88% | 6.04% | 5.23% | 5.96% |
| Radix-Spline | 10.19% | 1.64% | 3.56% | 8.96% | 6.09% |
| PGM | 16.76% | 2.2% | 1.07% | 21.29% | 10.33% |

## 4.3 Ablation Study of Dynamic $\epsilon$

To gain further insights about how the proposed dynamic $\epsilon$ framework works, we compare the proposed one with three dynamic $\epsilon$ variants: (1) *Random $\epsilon$* is a vanilla version that randomly choose $\epsilon$ from $[0, 2\tilde{\epsilon}]$ when learning each new segment; (2) *Polynomial Learner* differs our framework with another polynomial function $SegErr(\epsilon) = \theta_1 \epsilon^{\theta_2}$ where $\theta_1$ and $\theta_2$ are trainable parameters; (3) *Least Square Learner* differs our framework with an optimal (but very costly) strategy to learn $f(\epsilon, \mu, \sigma)$ with the least square regression.

Table 3: The AUSEC relative changes of dynamic $\epsilon$ variants compared to the proposed framework.

|  | Weblogs | IoT | Map | Lognormal | Average |
|---|---|---|---|---|---|
| Random $\epsilon$ | +70.94% | +68.19% | +51.73% | +73.38% | +66.06% |
| Polynomial Learner | +49.32% | +40.57% | +7.29% | +42.77% | +34.99% |
| Least Square Learner | +4.44% | +9.32% | +2.20% | −17.63% | −0.42% |

We summarize the AUSEC changes in percentage compared to the proposed framework in Table 3. Here we only report the results for FITing-Tree due to the space limitation and similar results can be observed for other methods. Recall that for AUSEC, the smaller, the better. From this table, we have the following observations: (1) The *Random $\epsilon$* version achieves much worse results than the proposed dynamic $\epsilon$ framework, showing the necessity and effectiveness of learning the impact of $\epsilon$. (2) The *Polynomial Learner* achieves better results than the *Random $\epsilon$* version while still have a large performance gap compared to our proposed framework. This indicates the usefulness of the derived theoretical results that link the index performance, the $\epsilon$ and the data characteristics together. (3) For the *Least Square Learner*, we can see that it achieves similar AUSEC results compared with the proposed framework. However, it has higher computational complexity and pays the cost of much larger building times, *e.g.*, $14\times$ and $50\times$ longer building times on IoT and Map respectively. These results demonstrate the effectiveness and efficiency of the proposed framework that adjusts $\epsilon$ based on the theoretical results, which will be validated next.

## 4.4 Theoretical Results Validation

We study the impact of $\epsilon$ on $SegErr_i$ for the MET algorithm in Theorem 1, where the derivations are based on the setting of the slope condition $a_i = 1/\mu$. To confirm that the proposed framework also works well with other $\epsilon$-bounded learned index methods, we analyze the learned slopes of other $\epsilon$-bounded methods in Appendix. In summary, we prove that for a segment $S_i : y = a_i x + b_i$ whose

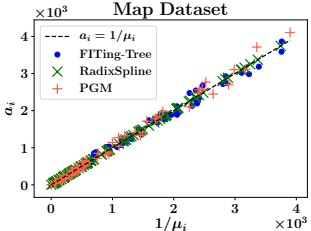
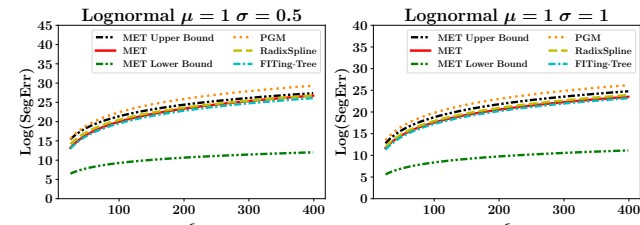

Figure 4: Learned slopes.      Figure 5: Illustration of the derived bounds.

covered data is $\mathcal{D}_i$ and the expected key interval of $\mathcal{D}_i$ is $\mu_i$, then $a_i$ concentrates on $1/\mu_i$ within $2\epsilon/(\mathbb{E}[Len(\mathcal{D}_i)]-1)$ relative deviations. Here we plot the learned slopes of baseline learned index methods in Figure 4. We can see that the learned slopes of other methods indeed center along the line $a_i = 1/\mu_i$, showing the close connections among these methods and confirming that the proposed framework can work well with other $\epsilon$-bounded learned index methods.

We further compare the theoretical bounds with the actual $SegErr_i$ for all the adopted learned index methods. In Figure 5, we only show the results on Lognormal dataset due to space limitation. As expected, we can see that the MET method has the actual $SegErr_i$ within the derived bounds, verifying the correctness of the Theorem 1. Besides, the other $\epsilon$-bounded methods show the same trends with the MET method, providing the evidence that these methods have the same mathematical forms as we derived, and thus the $\epsilon$-learner also works well with them.

## 4.5 CASE STUDY

We visualize the partial learned segments for FITing-Tree with fixed and dynamic $\epsilon$ on IoT dataset in Figure 6, where the $N$ and $\sum SegErr_i$ indicates the number of learned segments and the total prediction error for the shown segments respectively. The $\overrightarrow{\mu/\sigma}$ indicates the characteristics of covered data $\{\mathcal{D}_i\}$. We can see that our dynamic framework helps the learned index gain both smaller space (7 v.s. 4) and smaller total prediction errors (48017 v.s. 29854). Note that $\epsilon$s within $\overrightarrow{\epsilon_i}$ are diverse due to the diverse linearity of different data localities: For the data whose positions are within about $[30000, 30600]$ and $[34700, 35000]$, the proposed framework chooses large $\epsilon$s as their $\mu/\sigma$s are small, and by doing so, it achieves smaller $N$ than the fixed version by absorbing these non-linear localities; For the data at the middle part, they have clear linearity with large $\mu/\sigma$s, and thus the proposed framework adjusts $\epsilon$ as 19 and 10 that are smaller than 32 to achieve better precision. These experimental observations are consistent with our analysis in the paragraph under Eq. (2),

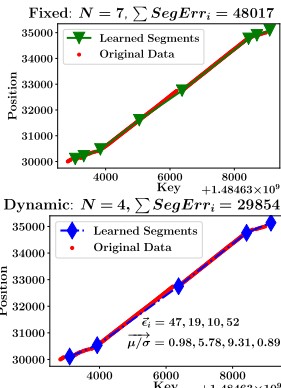

Figure 6: Visualization of the learned index (partial) on IoT for FITing-Tree with fixed $\epsilon = 32$ and dynamic version ($\tilde{\epsilon} = 32$).

and clearly confirm that the proposed framework adaptively adjusts $\epsilon$ based on data characteristics.

## 5 CONCLUSIONS

Existing learned index methods introduce an important hyper-parameter $\epsilon$ to provide a worst-case preciseness guarantee and meet various space-time user preferences. In this paper, we provide formal analyses about the relationships among $\epsilon$, data local characteristics and the introduced quantity $SegErr_i$ for each learned segment, which is the product of the number of covered keys and *MAE*, and thus embeds the space-error trade-off. Based on the derived mathematical relationships, we present a pluggable dynamic $\epsilon$ framework that leverages an $\epsilon$-learner to data-dependently adjust $\epsilon$ and achieve better index performance in terms of space-error trade-off. A series of experiments verify the effectiveness, efficiency and usability of the proposed framework.

We believe that our work contributes a deeper understanding of how the $\epsilon$ impacts the index performance, and enlightens the exploration of fine-grained trade-off adjustments by considering data local characteristics. Our study also opens several interesting future works. For example, we can apply the proposed framework to other problems in which the piece-wise approximation algorithms with fixed $\epsilon$ are used while still requiring space-error trade-off, such as similarity search and lossy compression for time series data (Chen et al., 2007; Xie et al., 2014; Buragohain et al., 2007; O'Rourke, 1981).

## REPRODUCIBILITY STATEMENT

For the Theorem 1 presented in Section 3.3, we give a complete proof in Appendix A. We introduce more implementation details of our method and baselines in Appendix E. More detailed description of the adopted datasets is included in Appendix F. To facilitate the reproducibiltity, we share downloadable source codes and IPython notebooks, and attach binary files of the public experimental datasets in the anonymous link [2].

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

APPENDICES FOR THE SUBMISSION: LEARNED INDEX WITH DYNAMIC $\epsilon$

## A  PROOF OF THEOREM 1

Given a learned segment $S_i : y = a_i x + b_i$, denote $c_i$ as the stored position of the last covered data for the $(i-1)$-th segment ($c_1 = 0$ for the first segment). We can write the expectation of $SegErr_i$ for the segment $S_i$ as the following form:

$$\mathbb{E}[SegErr_i] = \mathbb{E}\left[\sum_{j=0}^{(j^*-1)} |a_i X_j + b_i - (j + c_i + 1)|\right],$$

where $j^*$ indicates the length of the segment, and $X_j$ indicates the $j$-th key covered by the segment $S_i$. As studied in Ferragina et al. (2020), the linear-approximation problem with $\epsilon$ guarantee can be modeled as random walk processes. Specifically, $X_j = X_0 + \sum_{k=0}^{j} G_k$ (for $j \in \mathbb{Z}_{>0}$) where $G_k$ is the key increment variable whose mean and variance is $\mu$ and $\sigma^2$ respectively. Denote the $Z_j = X_j - j/a_i + (b_i - c_i - 1)/a_i$ as the $j$-th position of the transformed random walk $\{Z_j\}_{j \in \mathbb{N}}$, and $j^* = \max\{j \in \mathbb{N}| - \epsilon/a_i \leq Z_j \leq \epsilon/a_i\}$ as the random variable indicating the maximal position when the random walk is within the strip of boundary $\pm\epsilon/a_i$. The expectation can be rewritten as

$$\mathbb{E}\left[\sum_{j=0}^{(j^*-1)} |a_i X_j - j + (b_i - c_i - 1)|\right] = a_i \mathbb{E}\left[\sum_{j=0}^{(j^*-1)} |Z_j|\right]$$

$$= a_i \sum_{n=1}^{\infty} \mathbb{E}\left[\sum_{j=0}^{n-1} |Z_j|\right] \Pr(j^* = n). \tag{3}$$

The last equality in Eq. (3) is due to the definition of expectation. Following the MET algorithm that the $S_i$ goes through the point $(X_0, Y_0 = c_i + 1)$, we get $b_i = -a_i X_0 + c_i + 1$ and we can rewrite $Z_j$ as the following form:

$$Z_0 = 0, \quad Z_j \overset{j \geq 0}{=} X_j - X_0 - j/a_i = \sum_{k=1}^{j} G_k - j/a_i = \sum_{k=1}^{j}(G_k - 1/a_i) = \sum_{k=1}^{j}(W_k),$$

where $W_k$ is the walk increment variable of $Z_j$, $\mathbb{E}[W_k] = \mu - 1/a_i$ and $Var[W_k] = \sigma^2$. Under the MET algorithm setting where $a_i = 1/\mu$ and $\varepsilon \gg \sigma/\mu$, the transformed random walk $\{Z_j\}$ has increments with zero mean and variance $\sigma^2$, and many steps are necessary to reach the random walk boundary. With the Central Limit Theorem, we can assume that $Z_j$ follows the normal distribution with mean $\mu_{zj}$ and variance $\sigma_{zj}^2$, and thus $|Z_j|$ follows the folded normal distribution:

$$Z_j \sim \mathcal{N}\big((\mu - 1/a_i)j, j\sigma^2\big),$$
$$\mathbb{E}(|Z_j|) = \mu_{zj}[1 - 2\Phi(-\mu_{zj}/\sigma_{zj})] + \sigma_{zj}\sqrt{2/\pi}\exp(-\mu_{zj}^2/2\sigma_{zj}^2),$$

where $\Phi$ is the normal cumulative distribution function. For the MET algorithm, $a_i = 1/\mu$ and thus the $\mu_{zj} = 0$, $\sigma_{zj} = \sigma\sqrt{j}$, and $\mathbb{E}(|Z_j|) = \sqrt{2/\pi}\sigma\sqrt{j}$. Then the Eq. (3) can be written as

$$\frac{1}{\mu}\sum_{n=1}^{\infty}\mathbb{E}\left[\sum_{j=0}^{n-1}|Z_j|\right]\Pr(j^* = n) < \frac{1}{\mu}\sum_{n=1}^{\infty}\sum_{j=0}^{n-1}\mathbb{E}\left[|Z_j|\right]\Pr(j^* = n)$$

$$= \frac{\sigma}{\mu}\sqrt{\frac{2}{\pi}}\sum_{n=1}^{\infty}\sum_{j=0}^{n-1}\sqrt{j}\Pr(j^* = n). \tag{4}$$

For the inner sum term in Eq. (4), we have $(\sum_{j=0}^{n-1}\sqrt{j}) < \frac{2}{3}n\sqrt{n}$ since

$$\sum_{j=0}^{n-1}\sqrt{j} < \sum_{j=0}^{n-1}\sqrt{j} + \frac{\sqrt{n}}{2} < \int_0^n \sqrt{x}\,dx = \frac{2}{3}n\sqrt{n},$$

then the result in Eq. (4) becomes

$$\mathbb{E}[SegErr_i] < \frac{2}{3}\sqrt{\frac{2}{\pi}}\frac{\sigma}{\mu}\sum_{n=1}^{\infty}n\sqrt{n}\Pr(j^* = n)$$

$$= \frac{2}{3}\sqrt{\frac{2}{\pi}}\frac{\sigma}{\mu}\mathbb{E}[(j^*)^{\frac{3}{2}}] = \frac{2}{3}\sqrt{\frac{2}{\pi}}\frac{\sigma}{\mu}\mathbb{E}\left[((j^*)^2)^{\frac{3}{4}}\right] \leq \frac{2}{3}\sqrt{\frac{2}{\pi}}\frac{\sigma}{\mu}\left(\mathbb{E}[(j^*)^2]\right)^{\frac{3}{4}},$$

where the last inequality holds due to the Jensen inequality $\mathbb{E}[X^{\frac{3}{4}}] \leq (\mathbb{E}[X])^{\frac{3}{4}}$. Using $\mathbb{E}[j^*] = \frac{\mu^2}{\sigma^2}\epsilon^2$ and $Var[j^*] = \frac{2}{3}\frac{\mu^4}{\sigma^4}\epsilon^4$ derived in MET algorithm Ferragina et al. (2020), we get $\mathbb{E}[(j^*)^2] = \frac{5}{3}\frac{\mu^4}{\sigma^4}\epsilon^4$, which yields the following upper bound:

$$\mathbb{E}[SegErr_i] < \frac{2}{3}\sqrt{\frac{2}{\pi}}(\frac{5}{3})^{\frac{3}{4}}(\frac{\mu}{\sigma})^2\epsilon^3.$$

For the lower bound, applying the triangle inequality into the Eq. (3), we have

$$\frac{1}{\mu}\sum_{n=1}^{\infty}\mathbb{E}\left[\sum_{j=0}^{n-1}|Z_j|\right]\Pr(j^* = n) > \frac{1}{\mu}\sum_{n=1}^{\infty}\mathbb{E}\left[|\sum_{j=0}^{n-1}Z_j|\right]\Pr(j^* = n)$$

$$= \frac{1}{\mu}\sum_{n=1}^{\infty}\mathbb{E}[|Z|]\Pr(j^* = n), \tag{5}$$

where $Z = \sum_{j=0}^{n-1}Z_j$. Since $Z_j \sim N(0, \sigma_{zj}^2)$, the $Z$ follows the normal distribution:

$$Z \sim \mathbb{N}\left(\mu_Z = 0, \ \sigma_Z^2 = \sum_{j=0}^{n-1}\sigma_{zj}^2 + \sum_{j=0}^{n-1}\sum_{k=0, k\neq j}^{n-1}r_{jk}\sigma_{zj}\sigma_{zk}\right),$$

where $r_{jk}$ is the correlation between $Z_j$ and $Z_k$. Since $\mu_Z = 0$, the $|Z|$ follows the folded normal distribution with $\mathbb{E}[|Z|] = \sigma_Z\sqrt{2/\pi}$. Since the random walk $\{Z_j\}$ is a process with i.i.d. increments, the correlation $r_{jk} \geq 0$. With $\sigma_{zj} = \sigma\sqrt{j} > 0$ and $r_{jk} \geq 0$, we have

$$\mathbb{E}[|Z|] > \sqrt{\frac{2}{\pi}}\sum_{j=0}^{n-1}\sigma_{zj} > \sigma\sqrt{n(n-1)/\pi} > \frac{\sigma(n-1)}{\sqrt{\pi}},$$

and the result in Eq. (5) becomes:

$$\mathbb{E}[SegErr_i] > \frac{1}{\mu}\sum_{n=1}^{\infty}\mathbb{E}\left[|\sum_{j=0}^{n-1}Z_j|\right]\Pr(j^* = n)$$

$$> \frac{\sigma}{\mu}\sqrt{\frac{1}{\pi}}\sum_{n=1}^{\infty}(n-1)\Pr(j^* = n)$$

$$= \frac{\sigma}{\mu}\sqrt{\frac{1}{\pi}}\mathbb{E}[j^* - 1] = \sqrt{\frac{1}{\pi}}(\frac{\mu}{\sigma}\epsilon^2 - \frac{\sigma}{\mu}).$$

Since $\epsilon \gg \frac{\sigma}{\mu}$, we can omit the right term $\sqrt{\frac{1}{\pi}}\frac{\sigma}{\mu}$ and finish the proof.

## B    LEARNED SLOPES OF OTHER $\epsilon$-BOUNDED METHODS

As shown in Theorem 1, we have known how $\epsilon$ impacts the $SegErr_i$ of each segment learned by the MET algorithm, where the theoretical derivations largely rely on the slope condition $a_i = 1/\mu$. Here we prove that for other $\epsilon$-bounded methods, the learned slope of each segment (*i.e.*, $a_i$ of $S_i$) concentrates on the reciprocal of the expected key interval as shown in the following Theorem.

**Theorem 2.** *Given an $\epsilon \in \mathbb{Z}_{>1}$ and an $\epsilon$-bounded learned index algorithm $\mathcal{A}$. For a linear segment $S_i : y = a_i x + b_i$ learned by $\mathcal{A}$, denote its covered data and the number of covered keys as $\mathcal{D}_i$ and $Len(\mathcal{D}_i)$ respectively. Assuming the expected key interval of $\mathcal{D}_i$ is $\mu_i$, the learned slope $a_i$ concentrates on $\tilde{a} = 1/\mu_i$ with bounded relative difference:*

$$(1 - \frac{2\epsilon}{\mathbb{E}[Len(\mathcal{D}_i)] - 1})\tilde{a} \quad \leq \quad E[a_i] \quad \leq \quad (1 + \frac{2\epsilon}{\mathbb{E}[Len(\mathcal{D}_i)] - 1})\tilde{a}.$$

*Proof.* For the learned linear segment $S_i$, denote its first predicted position and last predicted position as $y'_0$ and $y'_n$ respectively, we have its slope $a_i = \frac{y'_n - y'_0}{x_n - x_0}$. Notice that $y_0 - \epsilon \leq y'_0 \leq y_0 + \epsilon$ and $y_n - \epsilon \leq y'_n \leq y_n + \epsilon$ due to the $\epsilon$ guarantee, we have $y_n - y_0 - 2\epsilon \leq y'_n - y'_0 \leq y_n - y_0 + 2\epsilon$ and the expectation of $a_i$ can be written as

$$\mathbb{E}[\frac{y_n - y_0 + 2\epsilon}{x_n - x_0}] \quad \leq \quad E[a_i] = \frac{y'_n - y'_0}{x_n - x_0} \quad \leq \quad \mathbb{E}[\frac{y_n - y_0 + 2\epsilon}{x_n - x_0}].$$

Note that for any learned segment $S_i$ whose first covered data is $(x_0, y_0)$ and last covered data is $(x_n, y_n)$, we have $\mathbb{E}[\frac{x_n - x_0}{y_n - y_0}] = \mu_i$ and thus the inequalities become

$$\frac{1}{\mu} - \mathbb{E}[\frac{2\epsilon}{x_n - x_0}] \quad \leq \quad E[a_i] \quad \leq \quad \frac{1}{\mu} + \mathbb{E}[\frac{2\epsilon}{x_n - x_0}].$$

Since $\tilde{a} = 1/\mu_i$ and $\mathbb{E}[x_n - x_0] = (\mathbb{E}[Len(\mathcal{D}_i)] - 1)\mu_i$, we finish the proof. $\square$

The Theorem 2 shows that the relative deviations between learned slope $a_i$ and $\tilde{a}$ are within $2\epsilon/(\mathbb{E}[Len(\mathcal{D}_i)] - 1)$. For the MET and PGM learned index methods, we have the following corollary that depicts preciser deviations without the expectation term $\mathbb{E}[Len(\mathcal{D}_i)]$.

**Corollary 2.1.** *For the MET method Ferragina et al. (2020) and the optimal $\epsilon$-bounded linear approximation method that learns the largest segment length used in PGM Ferragina & Vinciguerra (2020b), the slope relative differences are at $O(1/\epsilon)$.*

*Proof.* We note that the segment length of a learned segment is at $O(\epsilon^2)$ for the MET algorithm, which is proved in the Theorem 1 of Ferragina et al. (2020). Since PGM achieves the largest learned segment length that is larger than the one of the MET algorithm, we finish the proof. $\square$

## C   CONNECTING PREDICTION ERROR WITH SEARCHING STRATEGY

As we mentioned in Section 3.1, we can find the true position of the queried data point in $O(\log(N) + \log(|\hat{y} - y|))$ where $N$ is the number of learned segments and $|\hat{y} - y|$ is the absolute prediction error. A binary search or exponential search can be used to finds the stored true position $y$ based on $\hat{y}$. It is worth noting out that the searching cost in terms of searching range $|\hat{y} - y|$ of binary search strategy corresponds to the maximum absolute prediction error $\epsilon$, whereas the one of exponential search corresponds to the mean absolute prediction error (*MAE*). In this paper, we decouple the quantity $SegErr_i$ as the product of $Len(\mathcal{D}_i)$ and $MAE(\mathcal{D}_i|S_i)$ in the derivation of Theorem 1. Built upon the theoretical analysis, we adopt exponential search in experiments to better leverage the predictive models.

To clarify, let's consider a learned segment $S_i$ with its covered data $\mathcal{D}_i$. Denote the absolute prediction error of $k$-th data point covered by this segment as $|\hat{y_k} - y_k|$, the maximum absolute prediction error as $\epsilon_i$ where $|\hat{y_k} - y_k| \leq \epsilon_i$ for all $k \in [len(\mathcal{D}_i)]$.

- The binary search is conducted within the searching range $[\hat{y_k} \pm \epsilon_i]$ for each data point [3], thus the mean search range is $\sum_{k=1}^{len(\mathcal{D}_i)} \frac{1}{len(\mathcal{D}_i)} 2\epsilon_i = O(\epsilon_i)$, which is independent of the preciseness of the learned segment and an upper bound of $MAE(\mathcal{D}_i|S_i)$.

---

[3]The lower bound and upper bounds of searching ranges should be constricted to 0 and $len(\mathcal{D}_i)$ respectively. For brevity, we omit the corner cases when comparing these two searching strategies as they both need to handle the out-of-bounds scenario.

- The exponential search first finds the searching range where the queried data may present by centering around the $\hat{y}$, repeatedly doubling the range $[\hat{y} \pm 2^q]$ where the integer $q$ grows from 0, and comparing the queried data with the data points at positions $\hat{y} \pm 2^q$. After finding the specific range such that a $q_k$ satisfies $2^{\log(q_k)-1} \leq |\hat{y_k} - y_k| \leq 2^{\lceil \log(q_k) \rceil}$ for the $k$-th data, an binary search is conducted to find the exact location. In this way, the mean search range is $\sum_{k=1}^{len(\mathcal{D}_i)} \frac{1}{len(\mathcal{D}_i)} (2^{\lceil \log(q_k) \rceil + 1}) = O\big(MAE(\mathcal{D}_i|S_i)\big)$, which can be much smaller than $O(\epsilon_i)$ especially for strong predictive models and the datasets having clear linearity.

## D  THE ALGORITHM OF DYNAMIC $\epsilon$ ADJUSTMENT

---

***Algorithm***  *Dynamic $\epsilon$ Adjustment with Pluggable $\epsilon$ Learner*

---

**Input:** $\mathcal{D}$: *Data to be indexed,* $\mathcal{A}$: *Learned index algorithm,* $\tilde{\epsilon}$: *Expected $\epsilon$,* $\rho$: *Length percentage for look-ahead data*

**Output:** **S**: *Learned segments with varied $\epsilon$s*

1:  *initial parameters $w_{1,2,3}$ of the learned function:* $f(\epsilon, \mu, \sigma) = w_1(\frac{\mu}{\sigma})^{w_2} \tilde{\epsilon}^{w_3}$
2:  *initial mean length of learned segments so far:* $Len(\mathcal{D_S}) \leftarrow 404$
3:  $\mathbf{S} \leftarrow \varnothing$, $(\hat{\mu}/\hat{\sigma}) \leftarrow 0$
4:  **repeat**
5:      $(\mu/\sigma) \leftarrow lookahead(\mathcal{D}, Len(\mathcal{D_S}) \cdot \rho)$                       /* *get data statistic* */
6:      $\epsilon^* \leftarrow \left( \widetilde{SegErr} / w_1(\frac{\mu}{\sigma})^{w_2} \right)^{1/w_3}$         /* *adjust $\epsilon$ based on the learner* */
7:      $[\mathcal{S}_i, \mathcal{D}_i] \leftarrow \mathcal{A}(\mathcal{D}, \epsilon^*)$            /* *learn new segment $S_i$ using adjusted $\epsilon^*$* */
8:      $\mathbf{S} \leftarrow \mathbf{S} \cup \mathcal{S}_i$
9:      $\mathcal{D} \leftarrow \mathcal{D} \setminus \mathcal{D}_i$, $\mathcal{D_S} \leftarrow \mathcal{D_S} \cup \mathcal{D}_i$
10:     $Len(\mathcal{D_S}) \leftarrow running\text{-}mean\big(Len(\mathcal{D_S}), Len(\mathcal{D}_i)\big)$      /* *online update $Len(\mathcal{D_S})$* */
11:     $(\hat{\mu}/\hat{\sigma}) \leftarrow running\text{-}mean\big((\hat{\mu}/\hat{\sigma}), (\mu/\sigma)\big)$
12:     $w_{1,2,3} \leftarrow optimize(f, S_i, SegErr_i)$         /* *train the learner with ground-truth* */
13:     $\widetilde{SegErr} \leftarrow w_1(\hat{\mu}/\hat{\sigma})^{w_2} \tilde{\epsilon}^{w_3}$
14: **until** $\mathcal{D} = \varnothing$

---

In Section 3.4, we provide detailed description about the initialization and adjustment sub-procedures. The $lookahead()$ and $optimize()$ are in the **Look-ahead Data** and $\widetilde{SegErr}$ **and Optimization** paragraph respectively.

## E  IMPLEMENTATION DETAILS

All the experiments are conducted on a Linux server with an Intel Xeon Platinum 8163 2.50GHz CPU. We first introduce more details and the implementation of baseline learned index methods. *MET* (Ferragina et al., 2020) fixes the segment slope as the reciprocal of the expected key interval, and goes through the first available data point for each segment. *FITing-Tree* (Galakatos et al., 2019) adopts a greedy shrinking cone algorithm and the learned segments are organized with a B$^+$-tree. Here we use the stx::btree (v0.9) implementation (Bingmann, 2013) and set the filling factors of inner nodes and leaf nodes as $100\%$, *i.e.*, we adopt the full-paged filling manner. *Radix-Spline* (Kipf et al., 2020) adopts a greedy spline interpolating algorithm to learn spline points, and the learned spline segments are organized with a flat radix table. We set the number of radix bits as $r = 16$ for the Radix-Spline method, which means that the leveraged radix table contains $2^{16}$ entries. *PGM* (Ferragina & Vinciguerra, 2020b) adopts a convex hull based algorithm to achieve the minimum number of learned segments, and the segments can be organized with the help of binary search, CSS-Tree (Rao & Ross, 1999) and recursive structure. Here we implement the recursive version since it beats the other two variants in terms of indexing performance.

We then describe a few additional details of the proposed framework in terms of the $\epsilon$-learner initialization and the hyper-parameter setting. For the $w_{1,2,3}$ of the $\epsilon$-learner shown in the Eq. (2), at the beginning, we learn the first five segments with the $\epsilon$ sequence $[\frac{1}{4}\tilde{\epsilon}, \frac{1}{2}\tilde{\epsilon}, \tilde{\epsilon}, 2\tilde{\epsilon}, 4\tilde{\epsilon}]$, then track their rewarded $SegErr_i$ and update the parameters $w_{1,2,3}$ using least square regression. We empirically

found that this light-weight initialization leads to better index performance compared to the versions with random parameter initialization, and it benefits the exploration of diverse $\epsilon^*$, *i.e.*, leading to the larger variance of the dynamic $\epsilon$ sequence $[\epsilon_1, \ldots, \epsilon_i, \ldots, \epsilon_N]$. As for the hyper-parameter $\rho$ (described in the Section 3.4), we conduct grid search over $\rho \in [0.1, 0.4, 0.7, 1.0]$ on Map an IoT datasets. We found that all the $\rho$s achieve better space-error trade-off (*i.e.*, smaller AUSEC results) than the fixed $\epsilon$ versions. Since the setting $\rho = 0.4$ achieves averagely best results on the two datasets, we set $\rho$ to be $0.4$ for the other datasets.

## F  DATASET DETAILS

Our framework is verified on several widely adopted datasets having different data scales and distributions. *Weblogs* Kraska et al. (2018); Galakatos et al. (2019); Ferragina & Vinciguerra (2020b) contains about 715M log entries for the requests to a university web server and the keys are log timestamps. *IoT* Galakatos et al. (2019); Ferragina & Vinciguerra (2020b) contains about 26M event entries from different IoT sensors in a building and the keys are recording timestamps. *Map* dataset Kraska et al. (2018); Galakatos et al. (2019); Ding et al. (2020); Ferragina & Vinciguerra (2020b); Li et al. (2021) contains location coordinates of 1.8M places that are collected around the world from the Open Street Map OpenStreetMap contributors (2017), and the keys are compound by coordinates as $longitude + 90 \cdot latitude$. *Lognormal* Ferragina & Vinciguerra (2020b) is a synthetic dataset whose key intervals follow the lognormal distribution: $ln(G_i) \sim \mathcal{N}(\mu_{lg}, \sigma_{lg}^2)$. To simulate the varied data characteristics among different localities. We generate 20M keys with 40 partitions by setting $\mu_{lg} = 1$ and setting $\sigma_{lg}$ with a random number within $[0.1, 1]$ for each partition.

We normalize the positions of stored data into the range $[0, 1]$, and thus the key-position distribution can be modeled as Cumulative Distribution Function (CDF). We plot the CDFs and zoomed-in CDFs of experimental datasets in Figure 7 and Figure 8 respectively, which intuitively illustrate the diversity of the adopted datasets.

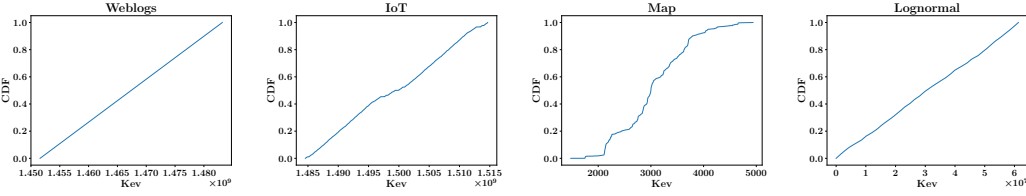

Figure 7: CDFs of adopted datasets.

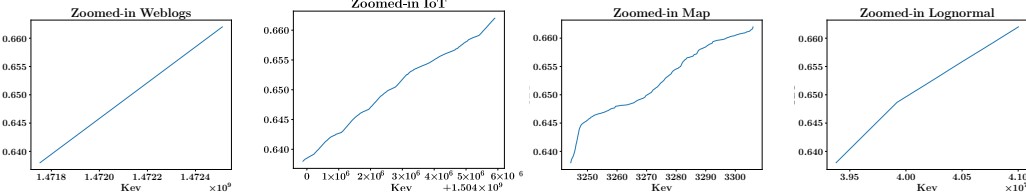

Figure 8: Zoomed-in CDFs of adopted datasets.

## G  ADDITIONAL EXPERIMENTAL RESULTS

**Overall Index Performance.**  For the space-error trade-off improvements and the actual querying efficiency improvements brought by the proposed framework, we illustrate more space-error trade-off curves in Figure 9 and querying time results in Figure 10. Recall that the $N$-*MAE* trade-off curve adequately reflects the *index size* and *querying time*: (1) the *segment size in bytes* and $N$ are only different by a constant factor, e.g., the size of a segment can be 128bit if it consists of two

double-precision float parameters (slope and intercept); (2) the querying operation can be done in $O(log(N) + log(|y - \hat{y}|))$ as we mentioned in Section 3.1, thus a better $N$-$MAE$ trade-off indicates a better querying efficiency. From these figures, we can see that the dynamic $\epsilon$ versions of all the baseline methods achieve better space-error trade-off and better querying efficiency, verifying the effectiveness and the wide applicability of the proposed framework.

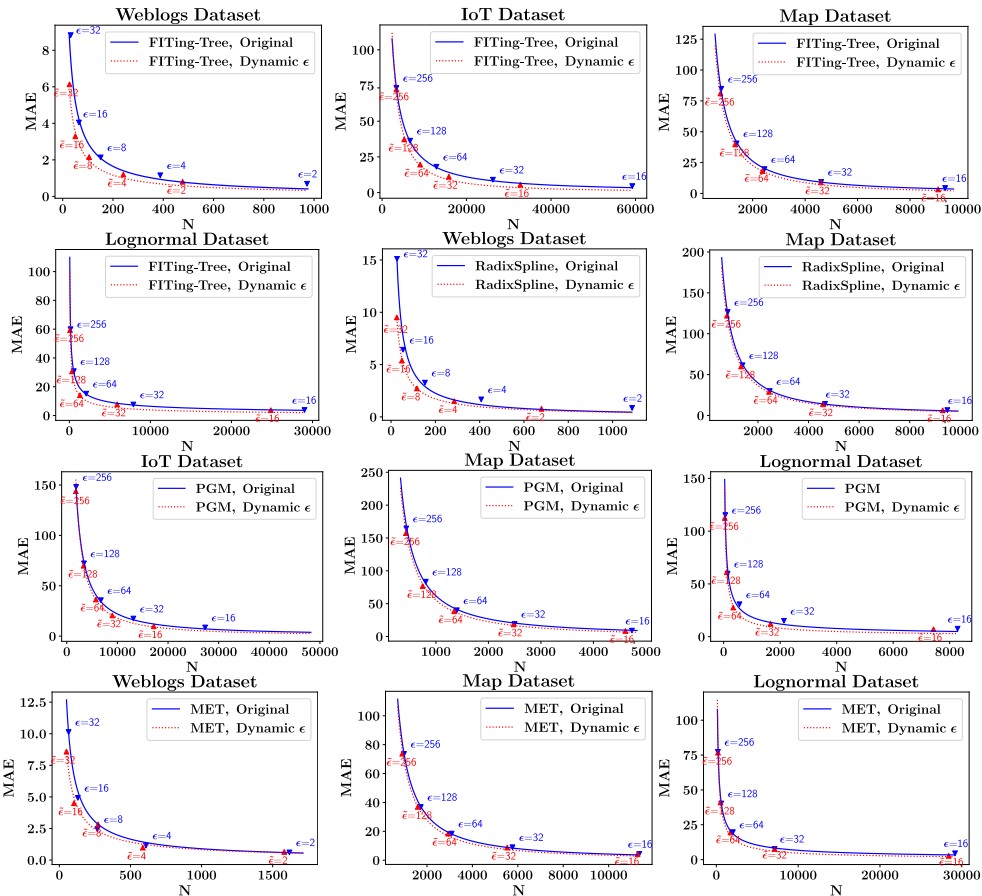

Figure 9: The additional space-error trade-off curves for learned index methods.

**Ablation Study.** To examine the necessity and the effectiveness of the proposed framework, in Section 4.3, we compare the proposed framework with three dynamic $\epsilon$ variants for the FITing-Tree method. Here we demonstrate the AUSEC relative changes for the Radix-Spline method with the same three variants in Table 4 and similar conclusions can be drawn.

Table 4: The AUSEC relative changes of dynamic $\epsilon$ variants compared to the Radix-Spline method with the proposed framework.

|  | Weblogs | IoT | Map | Lognormal | Average |
|---|---|---|---|---|---|
| Random $\epsilon$ | +81.23% | +74.78% | +59.20% | +83.16% | +74.59% |
| Polynomial Learner | +56.20% | +53.28% | +7.01% | +55.01% | +42.88% |
| Least Square Learner | -9.56% | +9.81% | +0.58% | $-11.23\%$ | $-2.60\%$ |

**Theoretical Validation.** In Section 4.4, we show that all the learned index baseline methods learn similar segment slopes on the Map dataset. Here we illustrate the learned slope results on the IoT, Weblogs and Lognormal datasets in Figure 11, which supports the Theorem 2 that the learned segment slopes concentrate on the $1/\mu_i$ with a bounded relative difference.

Besides, for the comparison between the theoretical bounds and the actual $SegErr_i$ of all the adopted learned index methods, we show more results on another two datasets *Gamma* and *Uniform* in Figure 12, where the key intervals of the two datasets follow gamma distribution and uniform distribution respectively. These results show that the MET method gains actual $SegErr_i$ within the bounds, verifying the correctness of the Theorem 1 again. Here all the learned index methods also achieve the same trends, showing that these methods have the same mathematical forms w.r.t. the $SegErr_i$, $\epsilon$ and $\mu/\sigma$, and hence the $\epsilon$-learner can effectively learn the estimator and adaptively choose suitable $\epsilon$.

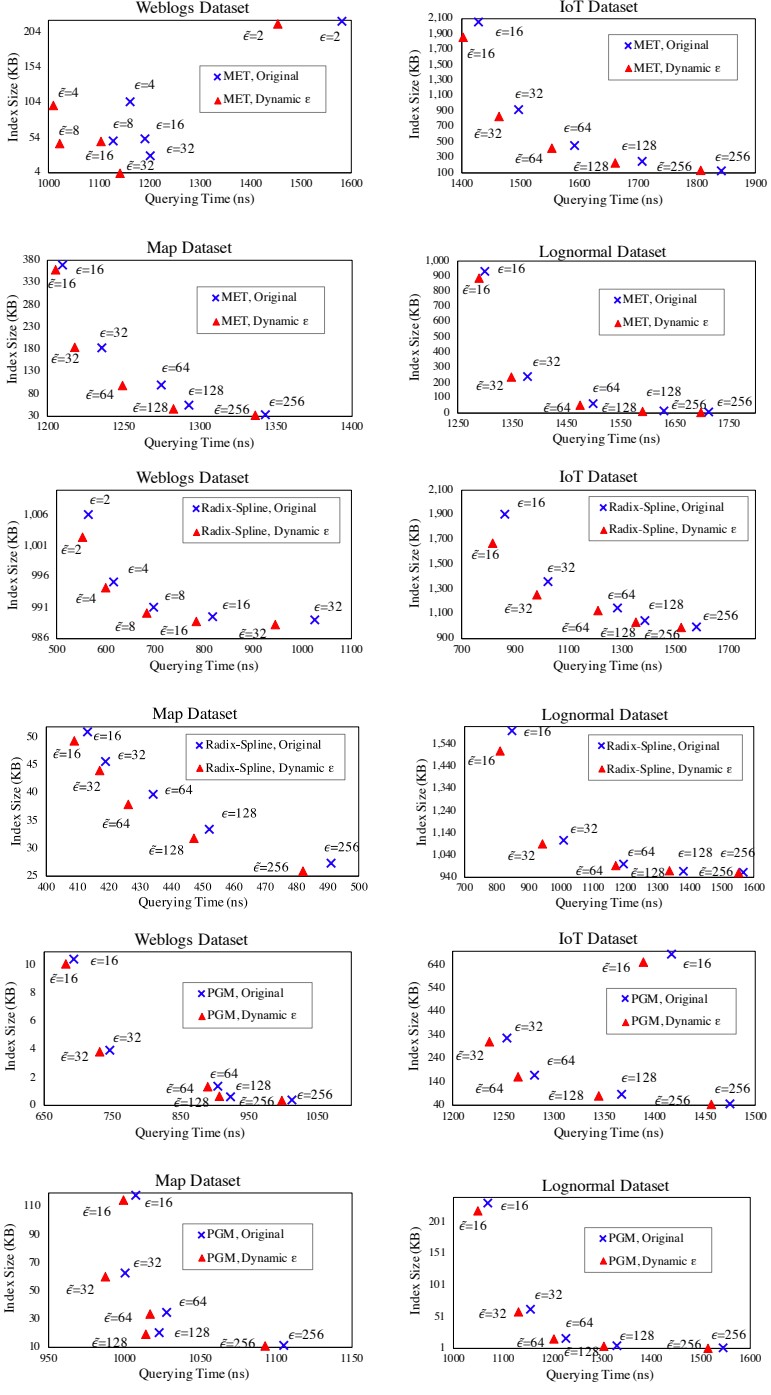

Figure 10: Improvements in terms of querying time for learned index methods with dynamic $\epsilon$.

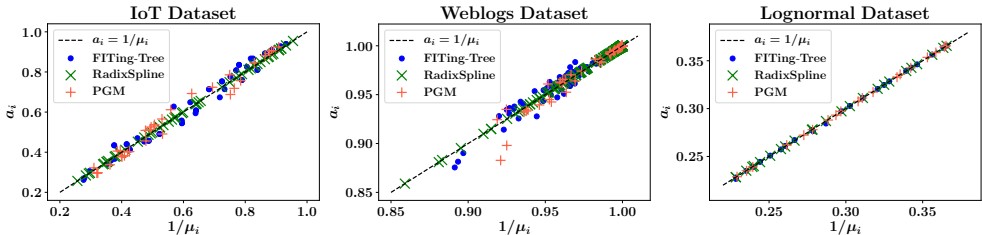

Figure 11: Learned slopes on the IoT, Weblogs and Lognormal datasets.

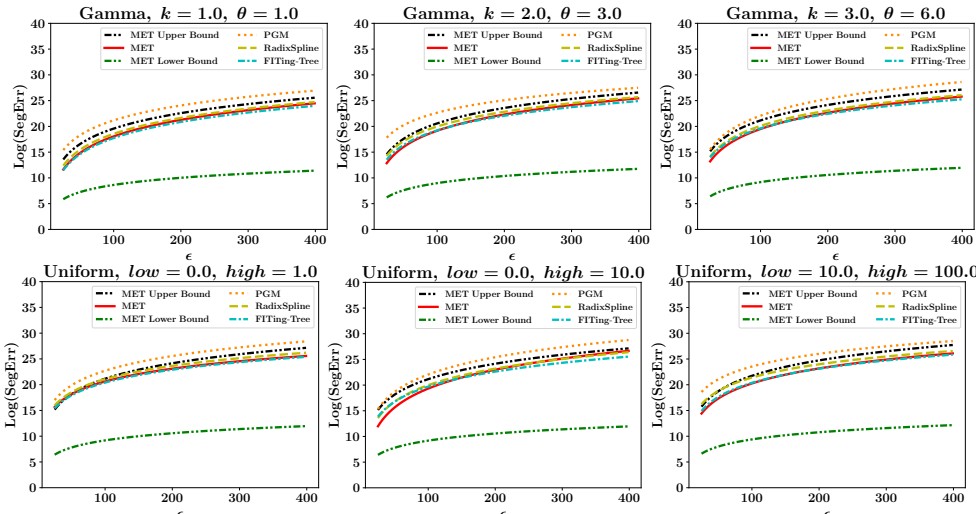

Figure 12: Illustrations of the derived bounds on *Gamma* and *Uniform* datasets.

