# OpenReview forum: "Learned Index with Dynamic $\epsilon$"
_ICLR.cc/2022/Conference — ICLR 2022 Submitted_

### Official Review · Reviewer_bkBs · 2021-11-03

**Correctness:** 3
**Technical Novelty And Significance:** 2
**Empirical Novelty And Significance:** 2
**Recommendation:** 8
**Confidence:** 3

**Main Review:**

Strengths:
1. The paper is well-written and easy to follow. I can easily understand the idea of this paper. The author first gives their intuition which motivates their proposed method.
2. This paper conducts extensive experiments on real-world datasets from different domains to show their improved performances, which is very reasonable and convincing to me.

Weaknesses:
1. There are some typos, in Table 3, the authors ignore the positive sign in the last column (compared to other columns),
2. The author can incorporate more explanations on the experimental results. For example, ‘From these results, we can see that the dynamic versions of all the baseline methods achieve much better error-space trade-off (( 16.48% to 23.57% averaged improvements as smaller AUSEC indicates better performance’, why there are some negative improvements?
3. There is one related reference, authors need to highlight their difference.

**Summary Of The Paper:**

This paper mainly studies the index structure problem.  Existing learned index methods use a fixed value for all the learned segments. In this paper, the author contributes a deeper understanding of how the impacts the index performance, and enlightens the exploration of fine-grained trade-off adjustments by considering data local characteristics.  Experiments with real-world datasets and several state-of-the-art methods demonstrate the efficiency, effectiveness, and usability of the proposed framework.

**Summary Of The Review:**

This paper is well-written and studies on the theoretical analysis of the database field, which focuses on an important and popular problem.

---

> ### Author Response · Authors · 2021-11-19
> **Reply to Review by Reviewer bkBs**
>
>
> Many thanks for your detailed comments and helpful suggestions! We appreciate that the submission receives your great recognition such as the paper is "well-written", and our work is "reasonable" and "convincing"!  We provide the following responses point by point for your comments:
>
> **"in Table 3, the authors ignore the positive sign in the last column (compared to other columns)"**
>
>    **Response**: Thank you for pointing out the missing positive sign. We add the sign accordingly in the revised version (the modification is marked in red).
>
> **"why there are some negative improvements?"**
>
>    **Response**: This is because that we evaluate the AUSEC metric and a smaller AUSEC indicates better indexing performance. The negative numbers mean that our method decreases AUESC and achieves a better trade-off. We explained this metric in detail in Section 4.1 as follows: "For a quantitative comparison w.r.t. the trade-off improvements, we calculate the area under the space-error curve (AUSEC) where the x-axis and y-axis indicate N and *MAE* respectively. For AUSEC metric, the smaller, the better."
>
> **"There is one related reference, authors need to highlight their difference."**
>
>    **Response**: We assume the mentioned "related reference" is MET [1], the most related work. The difference is two-folded as we described in Section 2.2.
>    From the view of theoretical analysis, [1] reveals the relationship between $\epsilon$ and index size performance, whereas our work analyzes the impact of $\epsilon$ on not only the index size, *but also the index preciseness*.
>    Besides, [1] only focuses on the theoretical analysis, while we leverage the theoretical results to dynamically adjust $\epsilon$ and improve indexing performance trade-off with the proposed pluggable framework. Thanks again!
>
>
> Reference
>
> [1] Ferragina, Paolo, Fabrizio Lillo, and Giorgio Vinciguerra. "Why Are Learned Indexes So Effective?." ICML, 2020.

---

### Official Review · Reviewer_it4h · 2021-11-04

**Correctness:** 3
**Technical Novelty And Significance:** 2
**Empirical Novelty And Significance:** 3
**Recommendation:** 3
**Confidence:** 3

**Main Review:**

The main strength of this paper is the empirical study, which shows some interesting results.  However, I am not sure that the theoretical justification is sound.  Namely, the main justification for using a dynamic $\epsilon$ seems flawed.  The authors argue that allowing for different $\epsilon$'s between segments allows the algorithm to decrease the number of linear segments at the cost of increasing the mean absolute error of some of the segments.  In the analysis of the runtime, it seems that what matters is the maximum absolute error in the predicted indexes, not the mean absolute error, which can be much smaller.  Thus it seems natural to me that prior methods guarantee that the maximum error of each segment is at most $\epsilon$.  Due to this, there seems to be a disconnect between the theoretical explanation of why the proposed method should work well and the method's improved empirical performance.  The proposed method seems to be able to trade-off between the number of segments and the mean absolute error within a segment, but not the maximum error which is what shows up in the runtime analysis.

Additionally, the overall presentation of the paper could be improved.  The paper could benefit from a more clear description of the $\epsilon$ tuning algorithm.  Related to the comments above, the motivation in section 3.1 could be improved by directly connecting the running time to the mean absolute error in a segment, as right now it seems to be connected to the maximum error.

Minor comments:
-  Use \log inside of math environments to represent logarithms

**Summary Of The Paper:**

This paper considers learning based methods for constructing index structures, a fundamental data structure.  Recent work has given learning based index structures which guarantee a maximum error of $\epsilon$ in the predicted index.  At a high level, this is done by breaking up the data set into several segments, and then a linear function is fitted to each segment, while maintaining the $\epsilon$ error guarantee.

The main idea behind this paper is to allow $\epsilon$ to vary across the different segments.  This is motivated by the fact that answering a query must first find the correct segment via a binary search, and then perform another binary search within the segment using the fitted linear function.  The runtime of the first step depends on the number of segments and the runtime of the second step depends on the prediction error.  By allowing the prediction error of some segments to go above $\epsilon$, the number of segments can be decreased, allowing for better trade-offs in some cases.  The main conceptual contribution is a method for tuning $\epsilon$ across different segments, which can be plugged directly into pre-existing learned index methods.    The authors give a theoretical analysis based on a stochastic analysis of the MET algorithm from prior work which is then leveraged to design their method for tuning $\epsilon$ across different segments.

An empirical study of their algorithm is performed on standard datasets for this area, comparing to several baselines from recent work in this area which use a fixed $\epsilon$.  The results show that the proposed $\epsilon$ tuning method can improve several learned index methods from prior work.

**Summary Of The Review:**

The paper provides some interesting experimental results on using dynamic $\epsilon$ in learned index structures, but the theoretical justification does not seem to explain the improved empirical performance adequately.

---

> ### Author Response · Authors · 2021-11-19
> **Reply to Review by Reviewer it4h**
>
> Thank you for your insightful suggestions! We are confident that we can address the issue raised. Our detailed responses are as follows:
>
> **[Theoretical Justification] "not sure that the theoretical justification is sound. In the analysis of the runtime, it seems that what matters is the maximum absolute error in the predicted indexes, not the mean absolute error, which can be much smaller. "**
>
>    **Response**: The theoretical justification is adequate to explain the improved empirical performance, due to the fact that we use the exponential search instead of binary search in the experiments.
>    As we mentioned in Section 3.1, we can find the true position of the queried data point in $O(\log(N) + \log(|\hat{y}-y|))$ where $N$ is the number of learned segments and $|\hat{y}-y|$ is the absolute prediction error. A binary search or exponential search can be used to find the stored true position $y$ based on $\hat{y}$.
> It is worth noting out that the *searching cost in terms of searching range* $|\hat{y}-y|$ of *binary search strategy* corresponds to the *maximum absolute prediction error $\epsilon$*, whereas the one of *exponential search* corresponds to the *mean absolute prediction error ($\textit{MAE}$)*. In this paper, we decouple the quantity $SegErr_i$ as the product of $Len(\mathcal{D}_i)$ and $\textit{MAE}(\mathcal{D}_i|S_i)$ in the derivation of Theorem 1. Built upon the theoretical analysis, we adopt exponential search in experiments to better leverage the predictive models.
>
> To clarify, let's consider a learned segment $S_i$ with its covered data $\mathcal{D}_i$. Denote the absolute prediction error of $k$-th data point covered by this segment as $|\hat{y_k} -y_k|$, the maximum absolute prediction error as $\epsilon_i$ where $|\hat{y_k} -y_k| \leq \epsilon_i$ for all $k \in [len(\mathcal{D}_i)]$.
>
> - The binary search is conducted within the searching range $[\hat{y_k} \pm \epsilon_i]$ for each data point, thus the mean search range is $\sum_{k=1}^{len(\mathcal{D}_i)} \frac{1}{len(\mathcal{D}_i)}2\epsilon_i = O(\epsilon_i)$, which is independent of the preciseness of the learned segment and an upper bound of $\textit{MAE}(\mathcal{D}_i|S_i)$.
>
> - The exponential search first finds the searching range where the queried data may present by centering around the $\hat{y}$, repeatedly doubling the range [$\hat{y} \pm 2^q$] where the integer $q$ grows from 0, and comparing the queried data with the data points at positions $\hat{y} \pm 2^q$. After finding the specific range such that a $q_k$ satisfies $2^{\log (q_k)-1} \leq |\hat{y_k} -y_k| \leq 2^{\lceil \log (q_k) \rceil}$ for the $k$-th data, an binary search is conducted to find the exact location. In this way, the mean search range is $\sum_{k=1}^{len(\mathcal{D}_i)} \frac{1}{len(\mathcal{D}_i)}(2^{\lceil \log (q_k) \rceil + 1}) = O\big(\textit{MAE}(\mathcal{D}_i|S_i)\big)$, which can be much smaller than $O(\epsilon_i)$ especially for strong predictive models and the datasets having clear linearity.
>
> In the revised version, we add the above clarification ("Appendix C. Connecting Prediction Error with Searching Strategy", marked in red), and refer readers to the direct connection between the running time and absolute prediction error in Section 3.1. Thanks again for helping us to improve the paper!
>
> **[Algorithm Description] "The paper could benefit from a more clear description of the $\epsilon$ tuning algorithm"**
>
>    **Response**: Thanks for your suggestion! We describe the $\epsilon$ tuning algorithm in section 3.4. For clarity, we add pseudo code in the revised version (Appendix D. The Dynamic $\epsilon$ Adjustment, marked in red). For your mentioned minor comment, we replaced the log into \\log accordingly.
>
> We are eager to work with the reviewer to make the submission clearer. We sincerely thank that the reviewer considers increasing the rating scores if our clarification addressed your comments. Please let us know if you have any additional questions or suggestions.

---

### Official Review · Reviewer_Fjr7 · 2021-11-07

**Correctness:** 2
**Technical Novelty And Significance:** 3
**Empirical Novelty And Significance:** 2
**Recommendation:** 3
**Confidence:** 4

**Details Of Ethics Concerns:**

No concern at this time.

**Main Review:**

The paper can be improved in terms of both argumentation and design.

The paper claim to resolve several tradeoffs at the same time through the proposed ε-learning approach:
It claims that the proposed method improves a space-time tradeoff in index performance.
At the same time, it claims to improve another trade-off, that of space-error.
Yet it in never clear how these three trade-off quantifies, space, time, and error are not compared to each other for a given ε value.
In particular, Figure 2 presents what appears to be a view of the space-error tradeoff. Yet it is not clear what ε values are set for each measured data point in the figure, and it is not clear what querying time they require. Perhaps improvement on one trade-off come at a cost on another.
Similarly, Figure 3 presents a good picture in terms of index size and querying time, yet it is not clear what cost that entails in terms of error.

On the design side, it is not clear why there should be user-define ε at all. If a user is interested on some index feature and presumably goes not wish to have a fixed ε guarantee, then it would be reasonable to do away with a user-specified ε, and use the proposed approach in order to guarantee some observable index feature, e.g., size, while adjusting ε as seems fit. The semantics of a dynamic ε guarantee from the user's point of view are unclear.

**Summary Of The Paper:**

This paper proposes a methodology, pluggable to any learned index method, that adjust the prediction error guarantee ε to data locality so as to achieve a given expected value of ε while improving index performance by some measure. Experiments with real world data show that the proposed approach can be plugged in several existing learned index method with improved results.

**Summary Of The Review:**

The paper resolves two trade-offs with clarifying the relationship among them, and does not explain the meaningfulness of a dynamic ε

---

> ### Author Response · Authors · 2021-11-19
> **Reply to Review by Reviewer Fjr7**
>
> Thanks for your positive comments and helpful suggestions! There is a little misunderstanding about the *trade-offs argumentation*, and the *usability of the introduced parameter $\epsilon$*. We have carefully gone through all your comments and give detailed responses as follows:
>
> **[Trade-offs Argumentation] "Yet it in never clear how these three trade-off quantifies, space, time, and error are not compared to each other for a given ε value."**
>
>    **Response**: Actually we propose four quantities and they appear in pairs. The first trade-off pair is the *number of learned segments ($N$)* and *mean absolute prediction error ($\textit{MAE}$)*. We put them together as they are directly derived from our theoretical analysis, and can be used to fairly reflect and examine the effectiveness of the derived bounds. The second pair is *index size* in KB and *querying time* in nanoseconds. We put them together because they are both quantities to represent the actual indexing performance for users.
>
>    Moreover, the improvements of *$N$-$\textit{MAE}$*  trade-off *adequately* reflects the improvements of the $\textit{Size}$-$\textit{Time}$ trade-off: (1) the *segment size in bytes* and $N$ are positively correlated and only different by a constant factor, e.g., the size of a segment can be 128bit if it consists of two double-precision float parameters (slope and intercept); (2) the querying time and $\textit{MAE}$ are also positively correlated. The querying operation can be done in $O(log(N)+log(|\hat{y}-y|))$ as we introduced in Section 3.1, thus a better *$N$-$\textit{MAE}$* trade-off indicates a better querying efficiency.
>
> **[Trade-offs Argumentation] "Perhaps improvement on one trade-off come at a cost on another"**
>
>    **Response**: For brevity, we did not mark the adopted $\epsilon$ and $\tilde{\epsilon}$ for each data points in Figure 2, in which we emphasize the comparison of AUESC metric. In fact, the curve in Figure 2 also contains the experiment points shown in Figure 3. Thanks for your question and **we mark these points in the new revised version (Figure 2 and Figure 9, the modifications are marked in red)**. From the results, we can see that *the proposed method indeed obtains two better trade-offs at the same time, due to the reason we stated above.*
>
>
> **[Usability of $\epsilon$] "it is not clear why there should be user-define $\epsilon$ at all"; "does not explain the meaningfulness of a dynamic $\epsilon$"; "If a user is interested on some index feature, ..., some observable index feature, e.g., size, ..."**
>
>    **Response**: The $\epsilon$ has been widely used in many existing learned index methods [1,2,3,4,5], and it is an intuitive, easy-to-set and method-agnostic quantity as we clarified in Section 3.2. On one hand, the prediction error can be linked to the memory accessing number in the querying process as $log|\hat{y}-y|$, and we can easily impose restrictions on the worst-case querying cases with $\epsilon$. On the other hand, compared to this quantity, the other quantities such as index size and querying time of a learned index model are harder to estimate, since they are dependent on specific algorithms, data layouts, implementations and experimental platforms.
>
> The proposed framework inherits the notion of $\epsilon$ from existing methods, retains the aforementioned benefits, and more importantly, provides users the same interface as the ones used by original learned index methods. That is, *we add no any additional cost to the users' experience, and users can smoothly and painlessly use our framework with given $\tilde{\epsilon}$ just as they use the original learned index methods with given $\epsilon$*. The $\tilde{\epsilon}$ is internally transformed to $SegErr$ and further leveraged to achieve a better performance trade-off, making the framework pluggable and easy to apply to a wide variety of existing learned index methods as we verified in experiments.
>
>    Regards to your mentioned "some observable index features, e.g., size", we note that the existing work PGM-index introduced a multi-criteria variant that auto-tunes itself with pre-defined "size" requirement. **Our proposed framework is pluggable and still valid when using the PGM variant to handle the "size" requirement**.
>    Specifically, given a space constraint, the multi-criteria PGM propose to iteratively estimate the relationship between $\epsilon$ and $size$ with a learnable function $size(\epsilon)=a\epsilon^{-b}$, and automatically outputs the index that minimizes its query time via different estimated $\epsilon$s.
>    Given a "size" requirement, we can just do the same thing in dynamic $\epsilon$ scene by setting our $\tilde{\epsilon}$ as $\epsilon$ estimated by the original PGM method.
>
> We sincerely thank that the reviewer considers increasing the rating scores if our clarification addressed your comments. Please let us know if you have any additional questions or suggestions.

---

> > ### Author Response · Authors · 2021-11-19
> > **Mentioned Reference in the Reply to Reviewer Fjr7**
> >
> > Reference
> >
> > [1] Galakatos, Alex, et al. "Fiting-tree: A data-aware index structure." SIGMOD, 2019.
> >
> > [2] Ferragina, Paolo, and Giorgio Vinciguerra. "The PGM-index: a fully-dynamic compressed learned index with provable worst-case bounds." VLDB, 2020.
> >
> > [3] Kipf, Andreas, et al. "RadixSpline: a single-pass learned index." aiDM, 2020.
> >
> > [4] Li, Pengfei, et al. "A scalable learned index scheme in storage systems." arXiv, 2019.
> >
> > [5] Ferragina, Paolo, Fabrizio Lillo, and Giorgio Vinciguerra. "Why Are Learned Indexes So Effective?." ICML, 2020.

---

### Author Response · Authors · 2021-11-26
**Gentle reminder of the final stage of discussion deadline**

Dear Reviewers,

We sincerely appreciate all the reviewers for your insightful and helpful comments! We do make great efforts during the author response period to clarify and address the comments from reviewers. For example, for the *trade-offs argumentation* mentioned by Reviewer Fjr7, we updated the experiments results with marked $\epsilon$ to illustrate how the method achieves two better trade-offs at the same time and make the argumentation more convincing; For the *mean/maximum prediction error in the theoretical justification* mentioned by Reviewer it4h, we present detailed theoretical connection between the two kinds of prediction error and specifc searching algorithms in the revision.

As the final stage of discussion comes to an end, we have received no responses. We would really appreciate it if you could kindly let us know there are any feedbacks or further questions. We are happy to engage more and address them fully!

Thanks, authors

---

### Decision · Program_Chairs · 2022-01-20

**Decision:**

Reject

**Comment:**

The authors present a new learning based algorithm for constructing index structure. Existing learned index algorithm use a fixed value, in contrast the authors show that a more refined methods can be used to obtain higher quality solutions for the problem.

The reviewers, after discussion, found the paper interesting and the experimental results promising but they feel that the paper in the current form is not yet ready for publication.  In particular,
- in the current form the theoretical motivation and the experimental results are a bit detached

Overall, the paper is interesting and the results are promising but it probably would benefit from significant re-writing before being accepted.